# Feasibility and Acceptability of Personalized Breast Cancer Screening (DECIDO Study): A Single-Arm Proof-of-Concept Trial

**DOI:** 10.3390/ijerph191610426

**Published:** 2022-08-21

**Authors:** Celmira Laza-Vásquez, Montserrat Martínez-Alonso, Carles Forné-Izquierdo, Jordi Vilaplana-Mayoral, Inés Cruz-Esteve, Isabel Sánchez-López, Mercè Reñé-Reñé, Cristina Cazorla-Sánchez, Marta Hernández-Andreu, Gisela Galindo-Ortego, Montserrat Llorens-Gabandé, Anna Pons-Rodríguez, Montserrat Rué

**Affiliations:** 1Department of Nursing and Physiotherapy and Health Care Research Group (GRECS), IRBLleida—Institut de Recerca Biomèdica de Lleida, University of Lleida, 25198 Lleida, Spain; 2IRBLleida—Institut de Recerca Biomèdica de Lleida, Department of Basic Medical Sciences, University of Lleida, 25198 Lleida, Spain; 3Department of Basic Medical Sciences, University of Lleida, 25198 Lleida, Spain; 4Heorfy Consulting, 25007 Lleida, Spain; 5Department of Computing and Industrial Engineering, University of Lleida, 25001 Lleida, Spain; 6Primer de Maig Basic Health Area, Catalan Institute of Health, 25003 Lleida, Spain; 7Proteomics and Genomics Service, University of Lleida, 25198 Lleida, Spain; 8Department of Radiology, Arnau de Vilanova University Hospital, 25198 Lleida, Spain; 9Breast Cancer Screening Program, Catalan Institute of Health, 25006 Lleida, Spain; 10Example Basic Health Area, Catalan Institute of Health, 25006 Lleida, Spain; 11Health PhD Program, University of Lleida, 25198 Lleida, Spain

**Keywords:** breast cancer, personalized screening, polygenic risk, risk assessment, acceptability, feasibility, proof-of-concept

## Abstract

The aim of this study was to assess the acceptability and feasibility of offering risk-based breast cancer screening and its integration into regular clinical practice. A single-arm proof-of-concept trial was conducted with a sample of 387 women aged 40–50 years residing in the city of Lleida (Spain). The study intervention consisted of breast cancer risk estimation, risk communication and screening recommendations, and a follow-up. A polygenic risk score with 83 single nucleotide polymorphisms was used to update the Breast Cancer Surveillance Consortium risk model and estimate the 5-year absolute risk of breast cancer. The women expressed a positive attitude towards varying the frequency of breast screening according to individual risk and, especially, more frequently inviting women at higher-than-average risk. A lower intensity screening for women at lower risk was not as welcome, although half of the participants would accept it. Knowledge of the benefits and harms of breast screening was low, especially with regard to false positives and overdiagnosis. The women expressed a high understanding of individual risk and screening recommendations. The participants’ intention to participate in risk-based screening and satisfaction at 1-year were very high.

## 1. Introduction

Personalized breast cancer screening is characterized by risk-based recommendations for the ages of screening initiation and cessation, the screening frequency, the type of screening exam, consideration of preventive treatments for high-risk women, or even decisions regarding the non-screening of low-risk women. In modeling studies, personalized risk-based approaches to the early detection of breast cancer appear to be more efficient and have a better balance of benefits and harms than age-only strategies [1,2,3,4,5].

Although personalized screening will probably be the standard for early detection in the near future, the systematic reviews warn of the need for additional studies that assess risk estimation, acceptability, feasibility, and the legal and ethical aspects of personalized screening strategies [3,4,5]. To respond to these needs, several initiatives are underway. Two non-inferiority clinical trials comparing risk-based screening with age-based screening—the Women Informed to Screen Depending on Measures of risk (WISDOM) [6] in the USA and My Personal Breast Screening (MyPeBS) [7] in Europe—will complete their respective collections of information by 2025. In Canada, the Personalized Risk Assessment for Prevention and Early Detection of Breast Cancer: Integration and Implementation (PERSPECTIVE I&I) project [8], and in the United Kingdom, the Predicting the Risk of Cancer at Screening (PROCAS) study [9], aim to improve personalized risk assessment, perform cost-effectiveness analyses, and identify best practices to implement them in their respective National Health Systems.

In 2019, the European Collaborative on Personalized Early Detection and Prevention of Breast Cancer (ENVISION) [10] organized a consensus conference with international consortia leading the research on personalized breast cancer screening, including the studies mentioned above. The consensus statement identified several areas of research that require development to enable evidence-based personalized breast cancer screening and prevention programs. These include breast cancer subtype-specific risk assessment tools and implementation studies addressing feasibility and acceptability combined with modeling studies to evaluate the long-term population outcomes of risk-based screening.

Personalized screening requires an accurate measure of individual risk. Age, reproductive history, breast density, family history of breast or ovarian cancer, previous benign breast disease, hormonal and lifestyle factors, and a combination of common genetic variants such as single-nucleotide polymorphisms (SNPs) and mutations in the BRCA or other susceptibility genes can be amalgamated to predict breast cancer risk [11,12,13,14,15]. Recently, the addition of artificial intelligence (AI) to breast-risk assessment—by means of digital imaging data obtained during the mammography screening process—has been proposed with promising preliminary results for short-term risk estimation [16,17].

A widely used risk model, the Breast Cancer Surveillance Consortium (BCSC) model developed and validated by Tice et al. in 2008 [18], includes age, race or ethnicity, breast density, family history of breast cancer, and previous biopsy. In 2015, an updated version of the BCSC model replaced the previous biopsy risk factor with the type of benign breast disease [12]. Yanes et al. showed that the addition of a polygenic risk score (PRS) to the existing risk models has improved their accuracy [19]. The highest AUC, 0.72, was reported in a model that combined BCSC, circulating estradiol levels, and PRS for estrogen receptor-positive breast cancer [20]. Hurson et al. [21] evaluated an integrated model incorporating classical risk factors and a 313-variant polygenic risk score to predict breast-cancer risk in fifteen prospective cohorts from six countries with more than 200,000 women of European ancestry. The authors concluded that integrating the 313-variant PRS with classical risk factors can improve the identification of women of European ancestry at an elevated risk who could benefit from targeted risk-reducing strategies under the current clinical guidelines.

In Spain, most population-based screening programs target women aged 50–69 and perform biennial mammograms. However, opportunistic screening in women younger than 50 years is widely used [22] and poses the potential harms of screening such as overdiagnosis and false positive results in low-risk women. Including women younger than 50 years in risk-based screening—so that low-risk women are recommended to wait, and thus high-risk women are screened—may improve the balance of benefits and harms of the intervention, as mathematical models have shown [2].

Studies assessing the acceptability of personalized screening by women who have participated in it are scarce. Although acceptability is generally good, the results are heterogeneous and vary according to a woman’s estimated risk of breast cancer. When less frequent screening is recommended, low-risk women are concerned about the possibility of a late diagnosis [23,24]. In contrast, in a European study, 94% of high-risk women positively rated increasing the frequency of screening [24]. When assessing the feasibility of implementing a personalized screening program, Alarie et al. [25] drew attention to concerns about the potential misuse of personal and genetic information by employers and insurers and highlighted women’s low awareness of the legislative framework for genetic discrimination. Women’s concerns about the potential costs associated with risk-based screening were raised by Rainey et al. [24], who proposed changes to the screening policy to ensure equal access, and by McWilliams et al. [23] and Rainey et al. [26], who identified women’s concerns about the need to improve the accuracy of breast cancer risk prediction and to provide personalized risk information.

In order to reduce the time lag between evidence generation and implementation, we designed the DECIDO project, which aims to assess the acceptability and feasibility of offering personalized breast cancer screening and its integration into the normal clinical practice. This proof-of-concept study, the protocol of which was published elsewhere [27], assessed women’s attitudes, participation intentions, and satisfaction with personalized breast screening.

## 2. Materials and Methods

For practical reasons, this section reproduces in part the methods section of the published protocol [27]. See the study protocol for more details.

### 2.1. Study Design

The study was designed as a single-arm proof-of-concept trial. A pilot study was carried out with 20 women to test the suitability of the recruitment and data collection processes and the coordination of the involved healthcare professionals.

### 2.2. Participants

From January 2019 to February 2021, 387 women aged 40 to 50 years were enrolled in the study. Potential participants were the 2038 women living in the “Primer de Maig” Basic Health Area in Lleida, Catalonia, on 31 December 2018, who would have turned between 40 to 50 years of age during the following 1.5 years. Accrual was suspended because of the COVID-19 pandemic in March 2020 when 252 women had been included and resumed in October 2020.

All women who turned 50 during the study period would have received the first invitation to participate in the population-based Breast Cancer Early Detection Program. Instead, they were invited to participate in our study. Women that declined were invited by the early detection program.

From women that turned 40 to 49 years during the study period, random samples of 20 to 50 women were selected from the potential participants on a monthly basis, and the women were invited to participate until the accrual goal was achieved.

Exclusion criteria included having a previous diagnosis of breast cancer, undergoing a current breast study, or fulfilling clinical criteria for cancer-related genetic counseling. We also excluded women not understanding or speaking Catalan or Spanish or those with a physical or cognitive disability that prevented breast screening or the main outcome’s assessment.

An invitation letter was mailed to the selected women, which was followed by a phone call 1–2 weeks afterwards where a healthcare professional of the study team provided a brief summary of the study and determined eligibility. Women that decided to participate were scheduled for a visit at the Primary Care center. Informed consent was obtained at the beginning of the first visit.

### 2.3. Intervention

The study intervention consisted of a baseline visit, the breast cancer risk estimation, a visit for risk communication and screening recommendations, the administration of a follow-up questionnaire, and a phone call to assess satisfaction after one year. Before the COVID-19 pandemic, all the visits were performed by physicians from the study team. As a result of the workload, during the pandemic, visits were made by a trained nurse, also a member of the study team, except for risk communication and recommendations to high-risk women. These visits continued to be performed by physicians.

The baseline visit was held at the Primary Care center, where the healthcare professional provided information about the study objectives; facilitated an informative brochure about the benefits and adverse effects of breast cancer screening [28]; obtained information on sociodemographic variables, risk factors, previous screening experience, perceived personal risk of breast cancer, and general screening knowledge, attitudes, and intentions; obtained a saliva sample to determine the genomic profile; and scheduled a screening mammogram with breast density measurement. For women that had a mammogram during the year before the first visit, breast density and presence/absence of benign lesions were obtained from that mammogram and the radiologist’s report.

Breast density was classified according to the Breast Imaging Reporting and Data System (BI-RADS), 5th edition, scoring system [29]: almost entirely fatty (a), scattered areas of fibroglandular density (b), heterogeneously dense (c), and extremely dense (d). Mammographic findings were coded from 0 (incomplete—additional imaging needed) to 6 (known biopsy—proven malignancy). In the case of abnormal results, additional tests were requested.

Collection, conservation, and delivery of saliva samples was completed following the saliva collection protocol provided by the University of Lleida’s Proteomics and Genomics Service. Details about the genotyping process can be found in the protocol [27]. The PRS was obtained using the 83 SNPs associated with breast cancer, based on Shieh et al.’s [30] or Mavaddat et al.’s [31] studies, as a composite likelihood ratio representing the individual effects of each SNP.

### 2.4. Breast Cancer Risk Estimation, Risk Communication, and Screening Recommendations

The Breast Cancer Surveillance Consortium v2.0 (BCSC v2.0) risk model [12] that includes age, race/ethnicity, first-degree relatives with breast cancer, history of benign breast disease, and mammographic breast density, together with the Catalan breast cancer incidence and mortality by causes of death and the risk factors distribution of the BCSC dataset, were used to estimate a preliminary 5-year absolute risk of breast cancer, as described by Gail et al. [32]. Then, a Bayesian approach was used to update the preliminary 5-year risk with the PRS value [27,30].

Once the breast cancer risk was obtained, a second visit was scheduled for risk communication and screening recommendations. The visit was performed by the same healthcare professional that performed the first visit. Women were informed of their 5-year absolute risk of breast cancer and the risk of a woman of the same age in the general population by means of two pictograms. Screening recommendations were given to them according to their risk, as described in the study protocol [27] and Appendix A in the Appendix A. Women with any anomalous or suspicious finding in mammography or with a very high risk of breast cancer after PRS assessment were referred to the public hospital Breast Unit. Since women 50 years old and older are invited biennially to breast cancer screening in the public system, there was no watch and wait recommendation for women of this age. Women that received recommendations to be screened annually or biennially and did not fulfill the age or periodicity criteria of the public-screening program were advised to inform their Primary Care doctors, who eventually could have referred them to the radiology unit for screening mammograms at the corresponding time points.

At the end of the risk communication visit, a follow-up questionnaire that included the primary and secondary outcomes was given to women. They were asked to return it within 2–4 weeks.

### 2.5. Outcomes

The primary outcome measures were attitude towards, intention to participate in, and satisfaction with personalized breast cancer screening by participating women.

Attitude was measured with a three-item scale [27,33], each item ranging from 1 to 5, with higher scores indicating more positive attitudes. A “positive attitude” was defined as a total score greater than or equal to 12. Intention to participate was measured with a 5-point Likert scale from definitely will (1) to definitely will not (5) [27,33]. The variable was also dichotomized as intending to participate (definitely or likely) or not. Satisfaction was assessed after one year of recruitment and was measured on a 5-point Likert scale from very unsatisfied (1) to very satisfied (5) [34].

Secondary outcomes (e.g., attitude towards screening mammography, attitude towards measuring breast cancer risk, emotional impact of the measure of breast cancer risk, preference with regard to the current screening, knowledge, decisional conflict, confidence, and participation) have been detailed in full in the study protocol [27].

### 2.6. Statistical Considerations

The sample size was estimated considering that the primary outcomes could be expressed as proportions that facilitate the interpretation as positive or neutral-negative outcomes. In order to ensure that the 95% confidence interval estimate of a proportion was within 5% of the true proportion, a sample of size 385 was needed.

Medians and quartiles or absolute and relative frequencies were used to describe quantitative and categorical variables, respectively. Likert scale responses corresponding to the primary outcomes were analyzed as ordered-categorical data with frequencies and proportions represented as stacked bar charts for the primary outcomes. The exact binomial test was used to obtain 95% confidence intervals for proportions. No imputation method was used when women omitted to supply answers to questions in the survey or the whole survey. The number of non-missing responses is always indicated.

The R programming language [35] and the RStudio environment [36] (R Foundation: Vienna, Austria) were used for the data analysis. The Likert function of the HH package [37] was used to obtain the graphical representation of the primary outcomes measured as Likert scales.

## 3. Results

From December 2018 to January 2021, 861 women were selected, and invitation letters were sent to them; of these selected women, 249 could not be reached and 38 did not meet the inclusion criteria (Figure 1). Of the remaining 574, 187 did not agree to participate. Thus, 387 women were included and gave informed consent, with an acceptance rate of 67.4%; of these women, 15 did not have an assessment of breast cancer risk (reasons specified in Figure 1), 14 withdrew from the study, 31 did not answer the Q2 questionnaire, and 12 did not answer the 1-year follow-up question on satisfaction. Therefore, 327 (84.5%) women answered all or part of the final questionnaire and 346 (89.4%) provided answers about their satisfaction with personalized screening around one year after inclusion.

### 3.1. Participants’ Characteristics

Table 1 describes the baseline characteristics of the participants. The median age was 48.4 years; two out of five have been invited to the population-based screening program for the first time, and two out of five had a university degree; four out of five were working; and three out of five had had mammograms previously. Three out of five women perceived their risk of being diagnosed with breast cancer during their lifetime as low or very low, and three out of five perceived their risk of being diagnosed with breast cancer the same as other women. The perceived knowledge of the benefits and harms of screening had medians of five and four, respectively, on a scale of one (very bad) to five (very good). Attitudes towards knowing the benefits and harms of breast cancer screening and risk-based screening were rated as “very much important” by more than half of the participants. Intentions of being involved in risk-based breast screening were rated as “very much right” and “very much important” for most of the women.

### 3.2. Risk of Breast Cancer and Screening Recommendations

The absolute risk of breast cancer at 5 years could be estimated for 372 (96.1%) women. Table 2 shows that 14% of the participants had a previous benign breast lesion, most of them unspecified. Breast density was classified as heterogeneously dense (4 out of 10 women) or extremely dense (1 out of 10) for more than half of the participants and 1 out of 10 reported a family history of breast cancer.

A median relative risk of breast cancer equal to 2.8 was obtained by comparing the risk characteristics of participants with the lowest risk categories of the BCSC v2.0 model. The median values of the estimated preliminary 5-year absolute risk of breast cancer and the PRS were 0.80% and 0.95, respectively. When the PRS was taken into account, the median of the updated 5-year absolute risk slightly decreased to 0.72% and the variability of the risk distribution increased, as the new quartiles and the plots of the distribution show (Figure 2).

One out of four women were classified as high risk and were recommended for annual screening (risk at 5 years > 1.16% for women aged 40–44 or > 1.19% otherwise). Only 14% (one out of seven) were recommended for biennial screening, one out of four triennial screening, and one out of three were recommended to watch and wait to be invited to the screening program at 50 years. For ages lower than 50 (before being invited to the screening program), two out of three were recommended to watch and wait and one out of four for annual screening. Three women had very high risk (> 6% at 5 years) and were referred to the hospital breast unit for a more intense follow-up.

Table 2 also includes the risk estimation components according to the screening recommendations. As expected, higher values of risk correspond to increased screening-frequency recommendations.

### 3.3. Primary Outcomes

Table 3 and Figure 3 display the primary outcomes. Attitudes towards personalized breast screening were positive, especially for varying the frequency of breast screening according to individual risk and for being invited more often if one’s breast cancer risk is higher than the average woman. Women expressed being less satisfied with being invited less often if they were found to have a lower risk of breast cancer. Overall, the attitude score was high, with a median of 12 on a scale of 3 to 15. Two out of three women met the threshold for a positive attitude (score > 12) towards personalized screening.

The intention to participate in personalized breast screening, if invited, was rated as “definitely will” or “likely to” by 9 out of 10 women. Around 1 year after being included in the study, 97% of the women declared they were satisfied or very satisfied with personalized breast screening and nine (2.6%) were not sure. Only one stated that she was very unsatisfied.

### 3.4. Secondary Outcomes

Table 4 Presents a summary of the secondary outcomes.

Attitudes towards undergoing breast screening (without specifying personalized screening) were positive overall. The median attitude score (on a scale of 5 to 25) was 22, and 9 out of 10 women met the threshold for a positive attitude (scores > 20). The attitude towards measuring breast cancer risk was also positive, with 3 out of 4 women considering that it would do more good than harm, in contrast to 1 out of 25 that expressed the opposite. More than three out of four women agreed that the information on individual breast cancer risk provided reassurance and more than one out of four agreed that receiving information about risks leads to anxiety or makes them worried. Compared to the current screening, two out of three would choose personalized screening and one out of four would choose the current screening.

Only 5 (1.5%) of 328 women had an adequate overall knowledge according to the established threshold for the three subscales of benefit, false positives, and overdiagnosis. An adequate knowledge of the three mentioned subscales was possessed by 18.1%, 6.5%, and 8.3% of the participants, respectively. For the conceptual knowledge items individually, 9 out of 10 women had correct answers for the definition of a screening mammogram, the benefit of screening towards the reduction of breast cancer deaths, and the possibility of having a false positive result. A low number of correct answers was given to two items on the overdiagnosis subscale: “screening causes some women with low-risk cancers, which would never cause any health problems, to be treated unnecessarily” and “screening is more likely to detect low-risk cancers, which would never cause any health problems, than to prevent death from breast cancer.” Both items were answered correctly by one out of four women. One out of two women answered (incorrectly) that the sentence “screening will not find every breast cancer” was false.

Scores for decisional conflict were low, with a median of 10 on a scale from 0 (no decisional conflict) to 100 (extreme decisional conflict). About 4 out of 10 women had a score equal to 0, which indicates no decisional conflict. Confidence in decision-making was high, with a median score of 4.7 on a scale from 1 to 5. The anxiety scores had a median of 30 on a scale from 20 to 80, with higher scores indicating greater levels of anxiety. Out of five women, one was quite worried or very worried about breast cancer, and the other four were split between being not worried at all or a bit worried. When deciding whether to have screening, avoiding death by breast cancer was mentioned as very important for more than 8 out of 10 women. Being diagnosed and treated for a cancer that is not harmful and having a false positive result were rated as very important for half of the participants.

Most women considered that being informed about breast cancer risk empowered them or increased their autonomy. Most of them expressed that receiving and commenting on risk information from a healthcare professional made them feel safer or better about making decisions that affected their health. Consistently with these results, the overall experience of having participated in the proof-of-concept study was assessed as positive by 8 out of 10 women. The confidence in personalized screening also received high rates with a median of five on a scale of one to five. Moreover, the understanding of the individual risk and the screening recommendations was rated very high. The median scores of the two items starting with *“I have understood…”* had the highest value (five = strongly agree) on a scale of one to five and the first quartile was four (agree) for both items. A total of 9 out of 10 women reported their intention (“definitely will” or “likely to”) to follow the recommendations.

The median time spent on risk communication was about 5 min, with 25% of the encounters with a time greater than or equal to 7 min.

## 4. Discussion

### 4.1. Main Results

This proof-of-concept study assessed the acceptability and feasibility of offering personalized breast cancer screening in the National Health System in Spain. The participating women expressed a positive attitude towards varying the frequency of breast screening according to individual risk and, especially, towards inviting women at higher risk more often than the average. A lower intensity screening for women at lower risk was not as welcomed, although half of the participants would accept it. The Intention to participate in risk-based screening and the satisfaction at one year were very high.

Concerning the risk measurement, the women expressed a high understanding of individual risk and screening recommendations. While a significant proportion of women stated that risk information could cause anxiety or worry, most had a positive attitude toward measuring risk and felt that it would do more good than harm.

Knowledge of the benefits and harms of breast screening was low, especially with regard to false positive results and overdiagnosis. Most women reported low decisional conflict, low or moderate levels of anxiety, and a high confidence in decision making. In deciding whether to have screening, the chance of avoiding death from breast cancer was considered very important more often than the chance of having a false positive result or being diagnosed and treated for a cancer that is not harmful. However, half of the women rated these adverse outcomes as very important.

### 4.2. Comparison with Other Studies

#### 4.2.1. Attitude towards Personalized Breast Cancer Screening and Intention to Participate

Recent studies have obtained results consistent with ours in relation to the acceptability of and intention to participate in personalized screening [24,38]. Dunlop et al. [38] explored the acceptability of risk-stratified screening in the Australian population across different cancer types. Almost all participants described tailored screening using personal-risk information as a positive and logical progression from the current one-size-fits-all approach and felt optimistic about using genomic information for estimating risk. In the PROCAS study, Evans et al. [39] investigated risk perception, the proportion wishing to know their 10-year risk, and whether subsequent screening attendance was affected. Women at high risk were more likely than those at low risk to perceive their risk as elevated prior to counseling, probably due to having a family history of breast cancer [24,39]. Both attendance at risk counseling and re-attendance at the subsequent mammography screening were higher in women counseled at high risk compared with those with low risk. Rainey et al. [24], based on 325 women from the PROCAS study, showed a high acceptability (94.8%) of screening recommendations. As the authors point out, risk feedback can motivate women for whom lifestyle interventions are likely to have the greatest benefit, that is, women with a high risk of breast cancer.

Additionally consistent with our findings, some studies have found that women are reluctant to reduce the frequency of screening or discontinue screening for low-risk groups, which healthcare professionals have mentioned as one of the barriers to the future implementation of the strategy [23,40,41]. Some women and health professionals have questioned whether breast cancer risk estimates could be accurate [40,41,42]. Others wonder whether it is financially motivated rather than an evidence-based safety measure [23,40,42,43,44]. As Dunlop et al. [38] remark, because of the high level of awareness socially reinforced by intense campaigns and media coverage, it may be challenging to reduce the breast cancer-screening frequencies in countries with established breast cancer screening programs. They suggest that offering a single screening episode with information about prevention and early detection, as opposed to no screening, may be more acceptable to those at low risk. This underlines the importance of comprehensive information materials and decision aids that outline the benefits and harms of all risk-tailored screening and preventive options.

Similar to other studies, our participants reported low levels of decisional conflict and anxiety about screening participation and a high confidence in decision making [22,33,45,46]. Smith et al. [46] found that both clinical and psychosocial factors influence cancer-screening decisions in older adults living in Australia. The reasons for continuing screening included perceived susceptibility to cancer, the value of knowing, needing reassurance, routine adherence, positive attitudes towards screening, and the benefits of screening. Long et al. [45], in a systematic review and thematic synthesis of qualitative studies on how women experience a false-positive test result from breast screening, found that women’s initial decisions to attend screening did not appear to be based on a prospective weighing of the harms and benefits of screening, but instead on a desire to substantiate their good health and to act conscientiously. In contrast, Hersch et al. [47], after exploring the potential mediators in the causal chain between exposure to over-detection information and subsequently reported breast screening intentions, reinforced the vital role of good educational materials and concluded that cognitions, rather than emotions, are instrumental in decision making.

Health professionals in Spain also stated the importance of general practitioners in the success of personalized breast screening [40]. Women’s trust in their primary care doctors, combined with a relatively high frequency of contact, would enable primary care professionals to monitor the follow-up of screening recommendations. Since a study with PROCAS participants [24] found that the concordance between self-reported and counseled risk was only 56.2%, Rainey et al. suggested the need to look to primary care professionals for assistance to take some of the responsibility away from women and aid women’s uptake and adherence to preventive measures.

#### 4.2.2. Breast Cancer Risk and Screening Recommendations

Most breast risk models estimate an absolute breast cancer risk over a specific timeframe, using personal, reproductive, and genetic risk factors. Clift el al. [48] point out the relevance of distinguishing between long-term risk prediction to guide screening strategies over a woman’s lifetime and estimating the risk of an underlying cancer being present at the time of screening. While a 5-year time frame is commonly used in models aimed at the general population, some authors propose using AI risk models with imaging data, over a 2-year timeframe, to identify women with a high probability of being diagnosed with a cancer that was missed, masked, or fast growing [16,17]. As Houssami et al. note that it is not clear whether the AI system’s short-term estimation is an enhanced early detection of existing cancers or a risk prediction [49].

Several studies have shown that the addition of PRS to the classical breast cancer risk factors improves risk stratification. Vachon et al. [50], based on the analysis of three case-control studies (1643 cases and 2397 controls) conducted in Germany and the USA, found that a 76-locus PRS and breast density were independent risk factors and that incorporating the PRS into the BCSC risk model improved the model fit and net reclassification for case patients. For instance, the PRS further stratified the risk associated with extremely dense breasts such that those with the lowest PRS had an odds ratio of 0.91, while those in the highest PRS had a 2.7-fold increased risk compared with women with scattered fibroglandular densities and average PRS. The BCSC-PRS model resulted in a net reclassification of 11% of cases exceeding the 5-year risk threshold of 3% or greater where chemoprevention should be discussed. Hurson et al. [21] evaluated an integrated model incorporating classical risk factors (without breast density) and a 313-variant PRS to predict breast cancer risk in 15 prospective cohorts from 6 countries. The integrated model, at the 3% 5-year risk threshold, reclassified 15% of US women, with around 6% moving from above the threshold to below and 9% moving in the opposite direction, leading to the identification of an additional 12% of future breast cancer cases. For the same threshold, the integrated model reclassified 15% of UK women (5% from above to below and 10% in the opposite direction), identifying an additional 16% of the future cases. Saghatchian et al. [51] reported the feasibility of giving screening recommendations based on individual risk in the general population. The use of PRS changed the risk score and monitoring and prevention recommendations in 40% of women, 28% of which classified from intermediate to moderate or high risk.

In our study, as has been described, we observed that the 5-year risk distribution showed increased variability after considering the PRS. Before the PRS, no woman had a 5-year risk above the 3% threshold, whereas after the PRS 15 (4%) women were reclassified above it. In addition, in terms of screening recommendations (annual or less intense), without considering the PRS, the recommendations would have been different for one in four women. Of those that would have been reclassified, 7 out of 10 would have been recommended for annual screening and the other 3 for less intense screening.

Since most PRS are based on data from individuals of a white European origin, they can produce inaccurate results in women of other ethnicities if differences in the allele frequencies or in the odds ratios for the disease association exist. Evans et al. [52] found that two well-validated PRS in the PROCAS study overestimated breast cancer risk, mostly for women of Black and Jewish origin. That indicates the need to develop PRS for women of non-White/non-European origin. In this sense, the results of the clinical trials that are currently being carried out (WISDOM, MyPeBS, PERSPECTIVE I&I, and PROCAS) that include large populations and various ethnic/racial groups could assess the calibration of risk models that include PRS.

#### 4.2.3. Risk Communication

A surprising result of our study is the high value that women placed in understanding the information they had received about their individual risks and their risk-based screening recommendations. Although the professionals that communicated risk in our study (two primary care doctors and two primary care resident doctors, in the pre-pandemic period, and one nurse when the study resumed during the COVID-19 pandemic) did not receive specific training for breast cancer risk communication in personalized screening, they had experience in communicating other risks in the clinical care setting and they were part of the research team that designed the DECIDO project. The supporting material used—consisting of a report including their statuses for family history, benign lesions, breast density, PRS, and two pictograms with the 5-year absolute risk of breast cancer, one with their risk and the other with the risk of a woman of the same age from the general population—was probably very helpful in the clinical encounter.

However, we found that conceptual and numerical knowledge of the benefits and harms of breast screening were low despite the fact that the study participants received an informative leaflet with detailed information on these benefits and harms. This result is consistent with a recent study on health literacy showing that one in every three to almost one in every two Europeans may not be able to understand essential health-related material [53]. In addition, there is evidence that both the general population and health professionals tend to have biased expectations of the benefits and harms of health interventions. Systematic reviews have shown that both women and health professionals overestimate the benefits and underestimate the harms of screening [54,55,56]. In addition, women overestimate the risk of having breast cancer and most of them have not been informed of the screening harms. The authors highlight that inaccurate perceptions about the benefits and harms of interventions are likely to result in suboptimal clinical management choices [55]. Strategies addressed to better engage women with limited health literacy undergoing mammography discussions are needed, including avoiding the assumption that mammographic procedures are common knowledge [57].

Pashayan et al. [10] indicate that the involved healthcare professionals need to have a clear understanding of the rationale for risk stratification and risk-tailored interventions, an adequate knowledge of screening-risk literacy and risk-communication skills, and they should also have access to structured referral pathways. Furthermore, these authors propose that risk-stratified interventions should be integrated across the continuum of training for healthcare professionals. Additionally, for women, as autonomy requires adequate knowledge and understanding to decide whether to follow the recommendations, it is essential to have adequate informational content, communication tools, and training for professionals.

In this regard, when Ahmed et al. [58] assessed the PERSPECTIVE e-platform directed at women in the general population of various literacy and education levels, they found that breast cancer risk and genetic testing were better understood by the participants. The authors concluded that such a platform may increase awareness and knowledge and support women’s informed decision making. Puzhko et al. [59] remarked on the importance of including training protocols in the original planning and design of a program’s implementation. They also pointed out that, for healthcare professionals, a major concern for personalized screening consists of time constraints. They considered it critical to allocate nurses or other trained personnel to help physicians explain the meaning and benefits of the program to the population. In addition, Ghanouni et al. [60] found that among women willing to have a risk assessment, letters/emails were generally preferred when the risk was low and face-to-face communication was most commonly preferred when the risk was high. General practitioners were the most preferred sources of assessment results and breast cancer specialists were often preferred for delivering the results of a very high-risk status. Blouin-Bougie et al. [61] highlight three main conditions that should be met to foster the acceptability of breast cancer risk stratification: respecting the principle of equity, paying special attention to knowledge management, and rethinking human resources to capitalize on the strengths of the current workforce.

### 4.3. Strengths and Limitations

This is the first proof-of-concept study that provides an indication that personalized breast cancer screening might be feasible and acceptable in the target population under the Spanish National Health System. The novelties of this study are (1) breast cancer-risk estimation using known risk factors and a polygenic risk score, (2) tailored recommendations on breast cancer screening provided to women by a primary care doctor or a nurse in their health center, and (3) the inclusion of women aged 40–49 years, an age interval not covered by the public screening program.

The study also has some limitations. First, the single-arm study design does not enable outcome comparisons between groups. However, this proof-of-concept study represents another step towards the implementation of personalized breast cancer screening in publicly funded health systems. Second, the results were based on a smaller sample compared to other published studies. Nonetheless, the study had a high acceptance and a low dropout rate, which ensures its validity and enables its generalization in the Spanish National Health System. Moreover, even though women that did not understand Catalan or Spanish were not included, the study sample was diverse with respect to age, educational level, and origin. Third, we did not explore the willingness to adopt preventive measures for women at high risk. Asking this type of question would have required providing additional information about the existing preventive interventions and about their pros and cons. Finally, the COVID-19 pandemic caused an interruption of around six months in the study’s accrual and development. It also caused a change in the risk communication process where a nurse of the study team took over for the primary care physicians who were at the healthcare frontline. The interruption in the study flow, for some women, may have caused non-response or recall problems when filling out the follow-up questionnaire or even study dropouts due to personal situations or a lack of interest. Finally, the low knowledge of the harms and benefits of screening for a non-negligible number of participants did not limit the understanding of the estimated risk and the screening recommendations.

## 5. Conclusions

This study provides information relevant to the development of risk-based breast screening in a publicly funded health system. The PRS introduced variability to the estimated breast cancer risk and modified the screening recommendations for a quarter of the sample. Participants had a positive attitude towards personalized breast screening; expressed their intention to participate in a personalized breast screening program, if possible; and were very satisfied with having participated in the study. However, women were relatively unwilling to accept less frequent screenings following the results of a low-risk status. The understanding of individual risk and the screening recommendations was high and most women expressed the intention to follow them.

## 6. Recommendations

Before implementing personalized screening, clear evidence of its benefits and harms must be established. In addition, for risk-based screening to be effective, valid risk models are needed, but equally important are the risk-based interventions that will be offered. In addition, more research is needed on its feasibility and acceptance by women through both quantitative studies with larger samples and diversity in the characteristics of women and qualitative studies that can deepen and broaden the scope of investigation, thereby enabling the realization of a better understanding of the factors that influence the acceptance of personalized screening by women.

Likewise, it is necessary to design tools and strategies for the development of interventions focused on raising awareness about personalized screening, increasing literacy on risk measurement, and achieving collaborative decision making in clinical practice.

## Figures and Tables

**Figure 1 ijerph-19-10426-f001:**
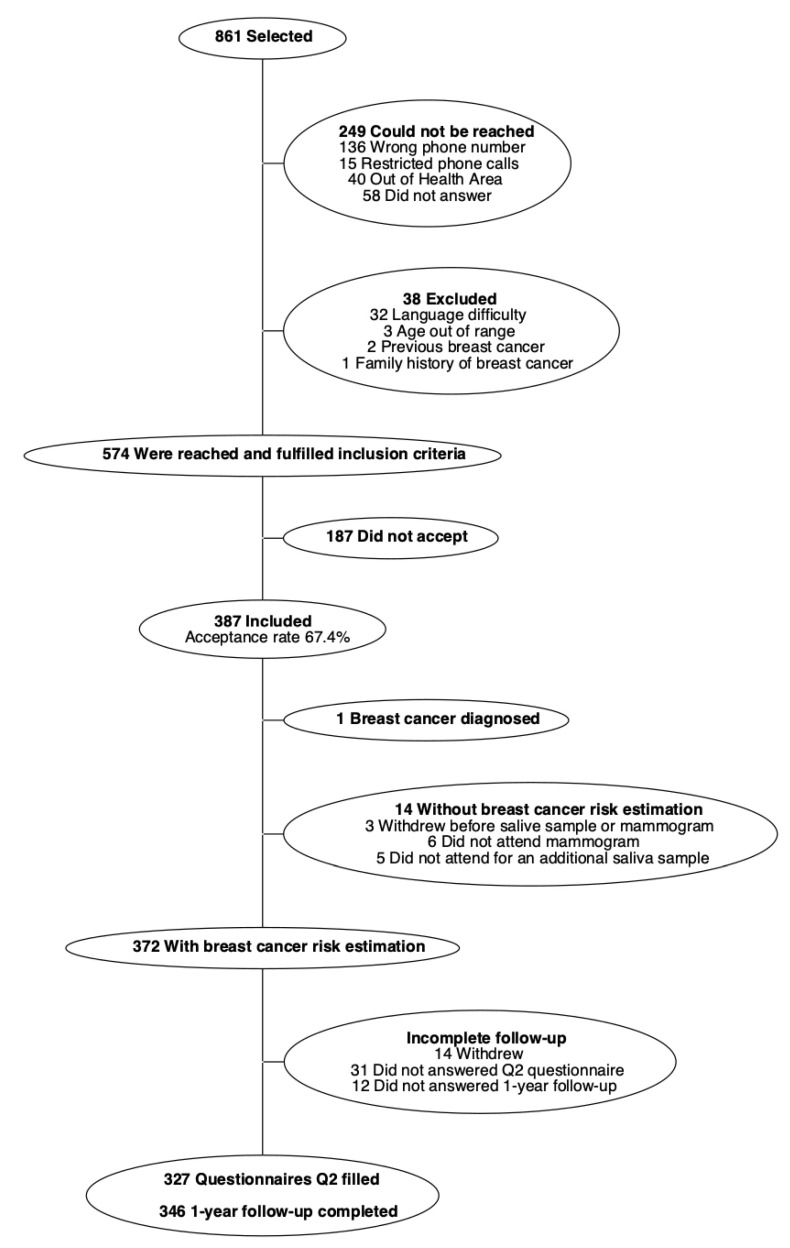
Study flowchart.

**Figure 2 ijerph-19-10426-f002:**
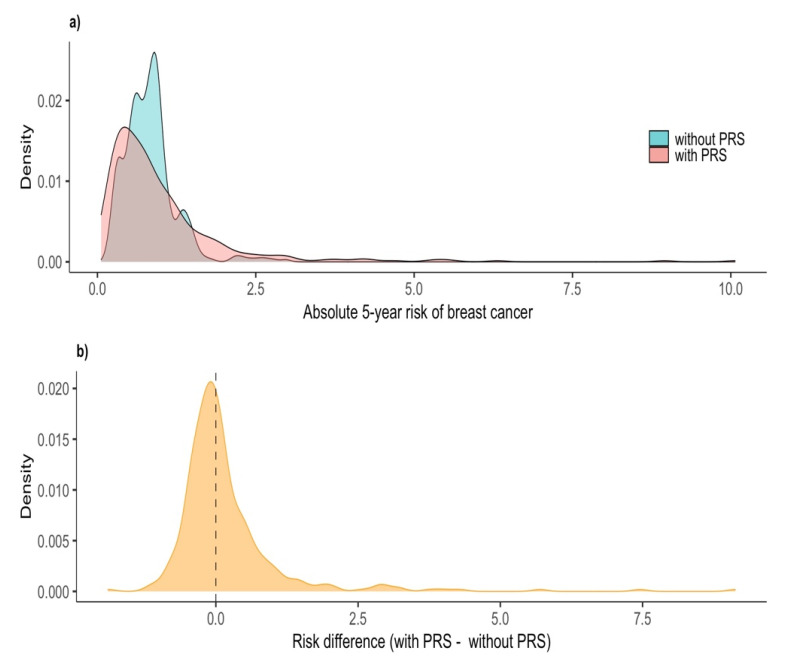
Absolute 5-year risk of breast cancer without (blue) and with (pink) the Polygenic Risk Score (PRS) in the risk model (**a**). Absolute 5-year risk difference, with PRS–without PRS (**b**).

**Figure 3 ijerph-19-10426-f003:**
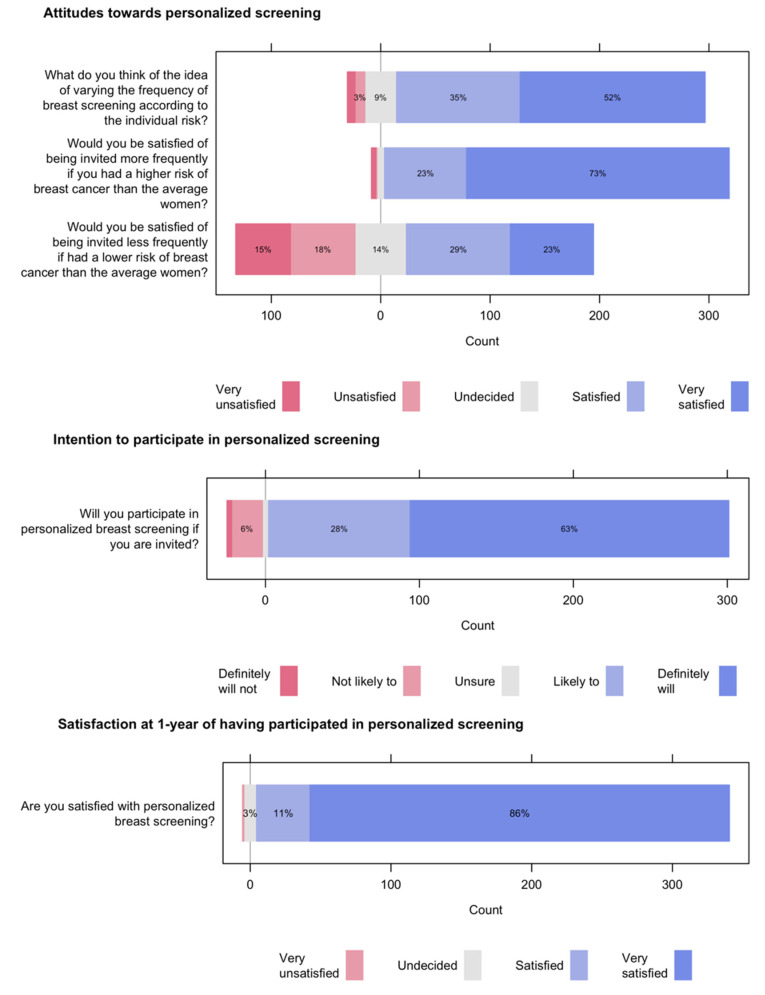
Likert scale distribution for the primary outcomes. (Note: Percentage values are rounded and values below 3% are not printed. Therefore, the sums may not add up to 100%.)

**Table 1 ijerph-19-10426-t001:** Baseline characteristics of participants.

Demographics and Health	Median (Q1; Q3) or *n* (%)	*n*
Age (years)	48.4 (43.8; 50.1)	387
Body mass index	25.4 (22.7; 29.4)	387
Source:		387
Primary care	211 (54.5%)	
Screening program	176 (45.5%)	
Birth place:		387
Catalonia	268 (69.3%)	
Other places in Spain	39 (10.1%)	
Other countries	80 (20.7%)	
Education:		387
Less than secondary school graduation	45 (11.6%)	
Secondary school diploma or equivalent	53 (13.7%)	
High school	125 (32.3%)	
University degree	164 (42.4%)	
Employment:		383
Working	306 (79.9%)	
No paid job	77 (20.1%)	
Had previous mammograms	218 (56.3%)	387
Age at first menstruation	13.0 (11.5; 14.0)	387
Number of children	2 (1; 2)	387
Age at first childbirth	28.0 (22.0; 32.0)	297
Breast feeding	192 (64.6%)	297
Peri- or postmenopausal	66 (17.1%)	387
Oral contraceptives		387
Never	116 (30.0%)	
<1 year	43 (11.1%)	
1–2 years	55 (14.2%)	
3+ years	173 (44.7%)	
**Perceived risk of being diagnosed with breast cancer during lifetime**		383
Very low	73 (19.1%)	
Low	160 (41.8%)	
Moderate	132 (34.5%)	
High	18 (4.7%)	
**Perceived risk of being diagnosed with breast cancer compared to other women**		382
Much lower	33 (8.64%)	
A little lower	63 (16.5%)	
The same	239 (62.6%)	
A little higher	46 (12.0%)	
Much higher	1 (0.3%)	
Perceived knowledge on benefits of breast screening	5 (4; 5)	384
Perceived knowledge on harms of breast screening	4 (3; 5)	380
**Attitudes towards knowing benefits and harms of breast screening and knowing risk-based screening** Not at all important (1) to very much important (5)		
For you, knowing the benefits is important	5 (5; 5)	384
For you, knowing the harms is important	5 (4; 5)	382
For you, knowing risk-based screening is important	5 (5; 5)	383
**General intentions of risk-based breast screening**		
For you, being involved in risk-based screening would be… Not at all right (1) to very much right (5)	5 (5; 5)	384
For you, being involved in risk-based screening would be… Not at all important (1) to very much important (5)	5 (5; 5)	377

**Table 2 ijerph-19-10426-t002:** Breast cancer risk estimation.

Variable	Median (Q1; Q3) or *n* (%)		*n*
**Previous benign lesion**			387
Non-proliferative disease	23 (5.9%)		
Unspecified benign lesion	31 (8.0%)		
None	333 (86.1%)		
**Breast density (BI-RADS)**			381
A: almost entirely fatty	56 (14.7%)		
B: scattered areas of fibroglandular density	117 (30.7%)		
C: heterogeneously dense	170 (44.6%)		
D: extremely dense	38 (10.0%)		
**Family history of breast cancer**	38 (9.8%)		387
**Risk estimation**			372
Relative risk ^a^	2.80 (1.85; 3.38)		
Absolute risk (%) (without PRS)	0.80 (0.58; 0.96)		
Polygenic risk score	0.95 (0.62; 1.51)		
Final risk (%) (with PRS)	0.72 (0.41; 1.19)		
Risk difference (with PRS minus without PRS, %)	−0.03 (−0.28; 0.34)		
**Recommendations ^b^**			
	Age < 50 years *n* = 204	Age 50 years *n* = 168	All *n* = 372
Referral to the hospital breast unit	1 (0.5%)	2 (1.2%)	3 (0.8%)
Annual screening	51 (25%)	39 (23.2%)	90 (24.2%)
Biennial screening	20 (9.8%)	32 (19.0%)	52 (14.0%)
Triennial screening	7 (3.4%)	95 (56.5%)	102 (27.4%)
Watch and wait until the screening program invitation	125 (61.3%)	0 (0.0%)	125 (33.6%)
**Risk estimation by screening recommendations**	Annual screening/referral to breast unit (*n* = 93)	Biennial screening (*n* = 52)	Triennial/wait until screening program invitation (*n* = 227)
Relative risk	3.35 (2.80; 4.03)	2.80 (1.85; 4.01)	2.08 (1.85; 2.96)
Absolute risk (%)	0.94 (0.82; 1.28)	0.92 (0.62; 1.16)	0.62 (0.48; 0.91)
Polygenic risk score	1.89 (1.44; 2.67)	1.13 (0.87; 1.56)	0.70 (0.50; 1.01)
Final risk (%)	1.73 (1.36; 2.48)	1.03 (0.89; 1.11)	0.46 (0.31; 0.67)
Risk difference (with PRS minus without PRS, %)	0.80 (0.47; 1.39)	0.12 (−0.15; 0.34)	−0.17 (−0.39; 0.00)

^a^: The relative risk for each woman was obtained by comparing her risk characteristics with the lowest risk categories of the BCSC v2.0 model. ^b^: Annual screening was recommended when the absolute risk of breast cancer at 5 years was higher than 1.16% for 40–44 years old women and 1.19%, otherwise. These risk values correspond to the average risks of 60 or 65-year-old women of the population, respectively.

**Table 3 ijerph-19-10426-t003:** Analysis of primary outcomes.

Primary Outcomes	Median (Q1; Q3) or *n* (%)	*n*
**Attitudes towards personalized breast screening** What do you think of the idea of varying the frequency of breast screening according to individual risk? Very bad idea (1) to very good idea (5)	5 (4; 5)	327
Would you be personally pleased to be invited more often for screening mammograms if you were found to have a higher risk of breast cancer than the average woman of your age? Not at all satisfied (1) to very satisfied (5)	5 (4; 5)	327
Would you be personally pleased to be invited less often for screening mammograms if you were found to have a lower risk of breast cancer than the average woman of your age? Not at all satisfied (1) to very satisfied (5)	4 (2; 4)	327
Overall attitudes score From 3 to 15; higher scores indicate more positive attitude	12 (11; 14)	327
Positive attitudes to personalized screening (scores ≥ 12) and 95% confidence interval	205 (62.7%)	327
	(57.2%, 67.9%)	
**Intentions to participate in personalized breast screening**		326
Intending to participate (definitely or likely)	299 (91.7%)	
95% confidence interval	(88.2%, 94.5%)	
Definitely will	207 (63.5%)	
Likely to	92 (28.2%)	
Unsure	20(6.1%)	
Not likely to or definitely will not	7 (2.2%)	
**Satisfaction with personalized screening (at 1 year)**	5 (5; 5)	346
Not at all satisfied	1 (0.3%)	
Dissatisfied	0 (0%)	
Neither satisfied nor dissatisfied	9 (2.60%)	
Satisfied	38 (11.0%)	
Extremely satisfied	298 (86.1%)	

**Table 4 ijerph-19-10426-t004:** Analysis of secondary outcomes.

Secondary Outcomes	Median (Q1; Q3) or *n* (%)	*n*
**Attitudes towards having breast screening**		
Overall attitudes score		
From 5 to 25; higher scores indicate more positive attitude	22 (21; 25)	286
Positive attitudes to screening (scores ≥ 20)	251 (87.8%)	286
**Based on what you know, do you think measuring breast cancer risk will do**		325
More good than harm	241 (74.2%)	
More harm than good	12 (3.7%)	
It depends	43 (13.2%)	
I don’t know	29 (8.9%)	
Information on the individual risk of breast cancer provides reassurance		
Strongly disagree (1) to strongly agree (5)	4 (4; 5)	327
Receiving information about risks leads to anxiety		
Strongly disagree (1) to strongly agree (5)	3 (2; 4)	327
The information about the individual risk of breast cancer makes me worried		
Strongly disagree (1) to strongly agree (5)	3 (2; 4)	326
**Preference with regard to the current screening** (biennial exams between 50 and 69 years)		326
Would choose personalized screening	215 (66.0%)	
Would choose the current screening	87 (26.7%)	
Don’t know	24 ( 7.3%)	
**Adequate knowledge (conceptual and numerical items combined)**		
Breast-cancer-mortality benefit	56 (18.1%)	310
False positives	21 (6.5%)	324
Overdiagnosis	24 (8.3%)	290
Adequate knowledge across all three subscales	5 (1.5%)	328
**Adequate knowledge (conceptual items individually)**		
Screening is for women without symptoms	283 (89.6%)	316
Screening reduces breast cancer deaths (benefit)	264 (95.7%)	276
Screening will not find every breast cancer (benefit)	130 (49.4%)	263
Screening may lead to false positive results (false positives)	275 (96.5%)	285
Screening increases breast cancer diagnoses (overdiagnosis)	225 (77.3%)	291
Overdiagnosis vs. false positives distinction (overdiagnosis)	101 (39.6%)	255
Not all breast cancers cause illness and death (overdiagnosis)	83 (34.3%)	242
Cannot predict if a cancer will cause harm (overdiagnosis)	182 (77.8%)	234
Cancer that might not cause problems is treated (overdiagnosis)	60 (25.3%)	237
Some women receive treatment they do not need (overdiagnosis)	92 (43.0%)	214
Screening overdiagnoses more often than prevents deaths (overdiagnosis)	60 (22.8%)	263
**Decisional conflict**		247
Score: no decisional conflict (0) to extreme decisional conflict (100)	10 (0; 25)	
0	93 (37.6%)	
1–24	78 (31.6%)	
≥25	76 (30.8%)	
**Confidence in decision making**		
Not at all confident (1) to very confident (5)	4.7 (4.0; 5.0)	322
Anxiety about screening participation		
Score, from 20 to 80, with higher scores indicating more anxiety	30 (20; 40)	297
Worry about breast cancer		308
Not worried at all	119 (38.6%)	
A bit worried	124 (40.3%)	
Quite worried or very worried	65 (21.1%)	
**Perceived significance of the benefits and the adverse effects of screening**		
In deciding whether to have screening, how important is it for you to consider the chance of…		
Avoiding breast cancer death		323
Very important	273 (84.5%)	
Quite important	32 (9.9%)	
A bit important	8 (2.5%)	
Not at all important	10 (3.1%)	
Being diagnosed and treated for a cancer that is not harmful		321
Very important	159 (49.5%)	
Quite important	90 (28.1%)	
A bit important	44 (13.7%)	
Not at all important	28 (8.7%)	
Having a false positive.		322
Very important	166 (51.6%)	
Quite important	83 (25.8%)	
A bit important	41 (12.7%)	
Not at all important	32 (9.9%)	
**Self-efficacy**		
Strongly disagree (1) to strongly agree (5)		
Breast cancer risk information makes me feel empowered as I have more knowledge	4 (3; 5)	322
Information about the risks of screening increases my autonomy	4 (3; 5)	322
Receiving and commenting on risk information from a healthcare professional makes me feel safer	5 (4; 5)	323
Receiving and commenting on risk information from a healthcare professional makes me feel better about making decisions that affect my health	5 (4; 5)	324
Overall self-efficacy score		
From 4 to 20; higher scores indicate higher self-efficacy.	17 (15; 19)	321
**Experience assessment**		
Overall experience score	22 (21; 25)	309
Positive experience to personalized screening (scores ≥ 20)	256 (82.8%)	
**Confidence in personalized screening** Not at all confident (1) to very confident (5)	5 (4; 5)	321
**Understanding of the individual risk and the screening recommendations**		
Strongly disagree (1) to strongly agree (5)		
I have understood the information I received about my risk for breast cancer in relation to women of my age	5 (4; 5)	319
I have understood the recommendations given to me about the screening of breast cancer in the coming years based on my risk of breast cancer	5 (4; 5)	320
**Intention to follow the recommendations about breast cancer screening**		321
Definitely will	244 (76.0%)	
Likely to	59 (18.4%)	
Unsure	11 (3.4%)	
Not likely to or definitely will not	7 (2.2%)	
**Time spent on risk communication** (minutes)	4.6 (3.2; 7.0)	370
**Proportion of women who agreed to participate in the study**	387 (67.4%)	574
**Proportion of participating women who completed the different phases of the study**		387
Risk estimation	372 (96.1%)	
Follow-up questionnaire	327 (84.5%)	
Assessment of satisfaction at 1-year	349 (90.2%)	

## Data Availability

The data presented in this study will be openly available in the Dryad repository at https://doi.org/10.5061/dryad.q83bk3jmc.

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
