# Peer review of "Feasibility and Acceptability of Personalized Breast Cancer Screening (DECIDO Study): A Single-Arm Proof-of-Concept Trial"

_ijerph, 2022, doi:10.3390/ijerph191610426_

Round 1

Reviewer 1 Report

The minor changes in language would make article more interesting read 

Author Response

Thank you very much. The article has been reviewed by a native English speaker.

Reviewer 2 Report

Review of the paper ”Feasibility and acceptability of ….” by Laza-Vasques et al.

This is a well written paper on a timely subject; what are women’s views on a move from age- to risk-based breast cancer screening? The authors have invited women aged 40 – 50 years to get their risk assessed, receiving recommendations for future screening and they have asked participants to answer surveys on a number of outcomes (e.g. knowledge of screening, views on risk based interventions). The authors cover the relevant literature, and the methods are sound.

General comments

Risk model

1.     The authors have chosen a 5-year risk model, probably because short-term models were not discussed when the study was planned. Risk based screening should preferably use a model with a time span as long as the screening interval of the program under study. That is, a model that focuses on identifying women who will develop interval cancers. There is no good reason for using a 5-year risk model in a biennial screening program if you assume that the risk is going to be assessed at each screening visit. 

2.     I also think that the authors should expand on the already long reference list. #19 includes representatives of a Swedish group that recently, 2020, published in Radiology. Eriksson et al showed that short term risk models, suitable for risk based screening, mainly relies on image based measures and that lifestyle factors and genetics does not add that much. Same goes for the models developed by the MIT group, Yala first author. In short, I would like a (short) discussion on what models are suitable to be used for risk based screening.

3.     Every here and there prevention is mentioned. For prevention maybe even life time models could be used. That different model should be used for different purposes should be mentioned. 

4.     Talking about prevention, did the participants get questions on willingness to adopt preventive measures if at high risk?

5.     Results could be better described, please see below under Tables/Figures

Specific comments

1.     Not clear to me why the study is described as a “single armed study”. This is not a trial.

2.     In the introduction the authors claim that personalised screening appears to more efficient and probably identifies cancers at lower stages. They also state that risk based screening is the future. If that was the case the Wisdom and MyPEBS trials would not be needed. We have absolutely no proof of that. Please soften that statement. 

3.     2.2 Participants. I have to admit I find the description of recruitment / inclusion a little hard to follow. For instance, how was the random selection of women aged 40-49 done? Maybe it is easier to separately describe the ones that were invited via the screening program and randomly?

4.     Line 188. What is meant with a watch and wait approach? Were high risk women, 40-49 years not invited for any follow up?

5.     Results. When calculating the attendance rate the denominator is 861. This was the number of women selected and invited. This means that 387/861 (45%) women were included, not 67%. 249 women could not be reached, what was the reason? 

6.     Table 1. 

a.     What does “Origin” mean (third line)?

b.     “Previous use of mammograms”Probably not the best way of phrasing it.

c.     “Menopause” should probably be changed to “Postmenopausal”

d.     Just giving the median and interquartile range (e.g. 5[5;5]) does not say much. Better to give the actual scores – how many gave a “not all important”, etc. 

7.     Table 2. 

a.     Please divide participants in high, moderate, general and low (or whatever you chose) followed by absolute and relative risk for each strata with the general risk as the reference (for the relative risk).

b.     Preferably you do this with and without the PRS. The “risk difference” is not intuitive and hard to interpret.

c.     “median relative risk” does not make sense. 

d.     In the text (line 274-280) I do not recognise the risk cut offs. According to the NICE guidelines a high risk is 8% 10 year risk, corresponding to a 4% 5 year risk. The authors use > 6% and end up with 3 (!) women at high risk. That is remarkably low. 

8.     Table 3

a.     As I understand it Table 3 and Figure 3 give more or less the same results, why?

9.     Table 4

a.     “Knowledge (conceptual items individually)”, what does the figures mean? For instance “Screening is for women without symptoms” = 89.6%. What does 89.6% mean?

10.  Discussion

a.     Well written and puts results in perspective. 

b.     That knowledge about the benefits and harms of screening is low is not surprising. Probably only marginally better among health care professionals. Are there any publications on the matter?

c.     Could the implications of “decisional conflict” be better described?

d.     Line 486. Be careful with the word “evidence”. You could have evidence once you have seen the recruitment rate of a full risk-based screening program. Your paper provides an indication that risk-based screening might be acceptable.

e.     I disagree with the authors when they state (line 520) that more research is needed on the feasibility and acceptance of risk-based screening before abandoning age based screening. I would argue that what is needed is proof of risk-based screening being more efficient than the one fits all approach. For risk-based screening to be effective valid risk models are needed, but equally important are the interventions. What are high risk women going to be offered? Ultrasound, MRI, contrast mammo, low dose tamoxifen, etc. What is needed are not studies on the acceptance but large studies challenging the old concept of age based screening and, hopefully, showing that risk based screening lowers overall and cause specific survival.

Reviewer 3 Report

This is a very interesting study assessing the feasibility of breast cancer risk assessment implementation. It also captured the attitudes of women enrolling in a precision screening program. The study is well done; the results show the benefits of a precision screening program as well as its limitations. The results could be enhanced by reporting the actual numbers and percentages (rather than 1 out of 5 or 1 out of 4 for example) especially since the attitudes are also reported on a likert scale.  The discussion thoroughly addresses the body of literature on this topic. The objectives of the study were met.

Author Response

We are glad that you found the study very interesting and well done. Thank you very much.

We appreciate your suggestion about the results section. However, since the tables already contain the absolute frequencies and percentages for most of the analyzed outcomes, we prefer to keep the wording of the text as it is to avoid redundancies and provide a better understanding and interpretation of the results.